# Structure and dynamics of an archetypal DNA nanoarchitecture revealed via cryo-EM and molecular dynamics simulations

Katya Ahmad [1,7], Abid Javed [2,6,7], Conor Lanphere [3], Peter V. Coveney [1,4,5] ✉, Elena V. Orlova [2] ✉ & Stefan Howorka [3] ✉

DNA can be folded into rationally designed, unique, and functional materials. To fully realise the potential of these DNA materials, a fundamental understanding of their structure and dynamics is necessary, both in simple solvents as well as more complex and diverse anisotropic environments. Here we analyse an archetypal six-duplex DNA nanoarchitecture with single-particle cryo-electron microscopy and molecular dynamics simulations in solvents of tunable ionic strength and within the anisotropic environment of biological membranes. Outside lipid bilayers, the six-duplex bundle lacks the designed symmetrical barrel-type architecture. Rather, duplexes are arranged in non-hexagonal fashion and are disorted to form a wider, less elongated structure. Insertion into lipid membranes, however, restores the anticipated barrel shape due to lateral duplex compression by the bilayer. The salt concentration has a drastic impact on the stability of the inserted barrel-shaped DNA nanopore given the tunable electrostatic repulsion between the negatively charged duplexes. By synergistically combining experiments and simulations, we increase fundamental understanding into the environment-dependent structural dynamics of a widely used nanoarchitecture. This insight will pave the way for future engineering and biosensing applications.

Nanostructures composed of DNA are of considerable interest as they can be folded via predictable DNA base-pairing into a wide range of geometries and sizes not easily accessible with other materials including biological proteins[1–4]. Reflecting these strengths, DNA nanostructures are used as research tools in materials science to position nanoparticles to study their nanoscale interaction[5], or in biophysics to shape lipid vesicles and control their fusion[6], but also for controlled drug delivery[7–9], nanorobotics[10–13], nanocomputing[14–17], and diagnostics[18–20]. DNA nanostructures can furthermore mimic several biological protein functions[1,2,21–26] including analyte binding[25,27–30],

motor-driven movement[31–35], and cytoskeletal structural support[6,36–39]. As an additional important topic in biomimicry, DNA nanostructures can replicate biological protein pores that puncture lipid bilayers and form conduits for transport across membranes[21,22,40–43].

Understanding the structure and dynamics of DNA nanoarchitectures is of fundamental interest and key for improving their rational design[44,45]. DNA nanostructures can be examined with atomic force microscopy and transmission electron microscopy[2,21,41,46–49]. These methods provide dimensions and structural information, but are limited by imaging resolution and represent only snapshots of the most

[1]Centre for Computational Science, University College London, London WC1H 0AJ, UK. [2]Department of Biological Sciences, Birkbeck, University of London, London WC1E 7HX, UK. [3]Department of Chemistry, Institute for Structural and Molecular Biology, University College London, London WC1H0AJ, UK. [4]Advanced Research Computing Centre, University College London, London WC1H 0AJ, UK. [5]Informatics Institute, University of Amsterdam, Amsterdam 1090 GH, The Netherlands. [6]Present address: Astbury Centre for Structural Molecular Biology, School of Molecular and Cellular Biology, Faculty of Biological Sciences, University of Leeds, Leeds, UK. [7]These authors contributed equally: Katya Ahmad, Abid Javed. ✉e-mail: p.v.coveney@ucl.ac.uk; e.orlova@mail.cryst.bbk.ac.uk; s.howorka@ucl.ac.uk

stable structures. To address the first shortcoming, high-resolution single particle static imaging[50] can be conducted, as shown recently for a few large DNA origami structures[3,51–53]. With regard to the second point, a better picture of dynamic structural changes can be obtained with modelling, including atomistic molecular dynamics simulations[54–60] and coarse-grained simulations[61–64] where atoms are grouped together to obtain longer simulation trajectories at a lower computational cost[65,66].

Several fundamental questions about DNA nanoarchitectures remain, however, unresolved largely due to the absence of any study that uses both high-resolution analysis and simulations. For instance, what is the relation between high-resolution structure and dynamics? Answering the question is key to improve the fundamental understanding of DNA nanoarchitectures and to maximise their biotechnological potential. The question should ideally be addressed with the underused ensemble mode which compensates for the often largely divergent individual simulation trajectories that often yield unreproducible data[67,68]. Ensemble averaging of trajectories[69] can substantially

reduce statistical uncertainty, making it much easier to perform direct comparisons between theory and experiment[68]. Another key question is how the structure and dynamics of DNA nanoarchitectures are influenced by anisotropic confinement posed by biologically important bilayer membranes[70]. In a biological example, the lipid bilayers of varying lateral membrane-pressure influence inserted mechanosensitive protein channels in terms of lumen width[71,72]. Analogous DNA nanostructures are hollow DNA barrels that insert into the anisotropic lipid bilayers to form transmembrane transport conduits[21,73]. An important question is if and how the DNA architectures differ outside and inside the membranes as any structural variation is expected to strongly influence the transport function.

Here we investigate the structural dynamics of an archetypal DNA motif in solution and the anisotropic environment of a bilayer membrane. The motif is the small barrel-like six-helix bundle (6HB) (Fig. 1a, Supplementary Tables 1 and 2)[41,42,73–77]. A bundle of six hexagonally arranged duplexes is widely used in DNA origami[21,52,78–83] including DNA nanotechnology design software[41,84]. Furthermore, the hollow 6HB

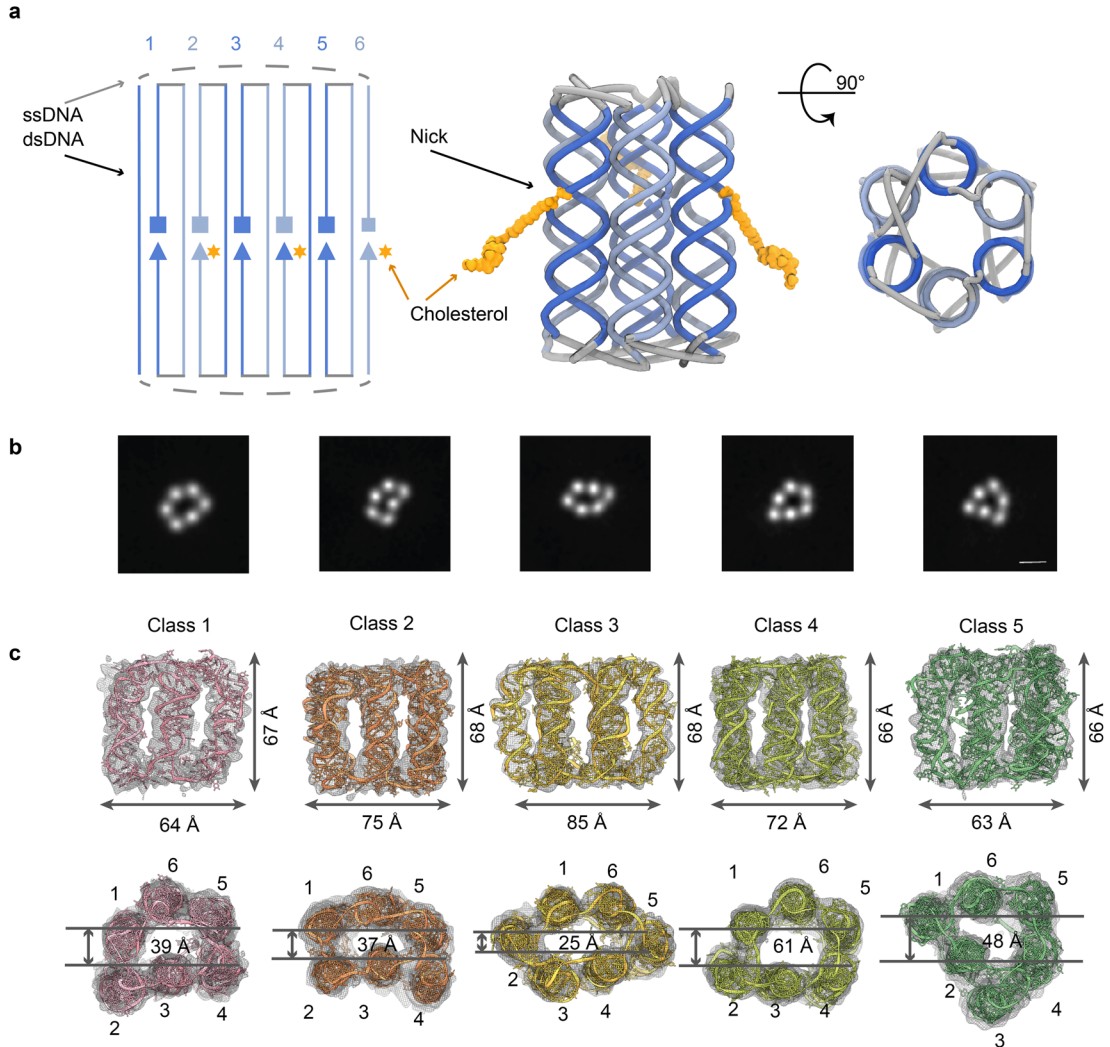

**Fig. 1 | Structural analysis of the small six-helical DNA bundle 6HB using cryo-EM imaging. a** 2D strand map and nominal 3D representation of 6HB. Each duplex is composed of 21 base-pairs connected at the top and bottom by ssDNA polythymidine loops. The DNA duplexes are numbered and shown in alternating dark and light blue. In the 2D strand map, the 5′ end of each strand is indicated by a square and the 3′ end by a triangle. The cholesterol modifications for inserting 6HB into membranes are indicated by asterisks (orange). The 3D representation shows the cholesterol lipid anchors attached via tri(ethylene glycol) linkers to the 3′ end of the DNA strands. **b** Cryo-electron microscopy (cryo-EM) 2D class averages of 6HB single-particle images, showing five different conformations. The scale bar corresponds to 20 Å. **c** Cryo-EM maps of the five conformational classes of 6HB shown as an overlay of EM map on the fitted atomic models with indicated dimensions. The bottom panel shows views from the top of the EM maps with fitted atomic models, and measured widths of 6HB. The conformational classes are represented in different colours.

barrel can be engineered with membrane anchors to span lipid bilayers (Fig. 1a)[41,85]. This makes it ideal to investigate how anisotropic membrane environments affect the structure and dynamics of DNA motifs. Furthermore, bilayer-spanning hollow DNA nanopores are of interest in biomimicry[40,41,47,86–88] and their utility for sensing[21,43,47,89], synthetic biology[42,90], drug delivery[22,74,75,88], targeted cell killing[22,87], and biophysical research[42,58,74,75,91–93].

To gain key insight into the structural dynamics of the archetypal 6HB (Fig. 1a)[42,74–76,94] we use cryo-electron microscopy (cryo-EM) and molecular dynamics simulations in an integrated synergistic approach. We first assess the 6HB structure in solution using cryo-EM. Our analysis confirms that the duplex bundle correctly assembles but exhibits many dynamic structures, different from the original cylindrical barrel-like design. The lumen of 6HB also considerably deviates from the designed cylindrical shape due to duplex distortion. Our simulations corroborate the structure from cryo-EM and provide insight into the dynamics and pore stability in different, experimentally important salt conditions, as well as the molecular basis of the structural dynamics such as duplex deformation. Compared to 6HB in solution, simulations of 6HB inserted into lipid bilayer reveal a drastic change to the designed cylindrical structure. This change is due to the lateral membrane pressure which pushes the duplexes into the designed parallel hexagonal alignment. Our study further simulates the flow of solvated ions through membrane-inserted barrels and achieves good agreement with experiments, thereby validating our simulations.

## Results

### Cryo-EM imaging

We used cryo-EM to investigate the structure of 6HB. This DNA nanoarchitecture is composed of six DNA duplexes assembled from six oligonucleotides (Fig. 1a)[41,42,74–76]. Each duplex folds from two DNA strands which connect to two neighbouring duplexes via interduplex loops (Fig. 1a). Each duplex features a single-stranded nick region in the barrel midsection (Fig. 1a). At the nick region of three duplexes, a cholesterol membrane anchor is attached to the 3′ end of the DNA strands (Fig. 1a). The elongated 6HB structure has a nominal height of ~90 Å with nominal outer and inner channel widths of 50 and 18 Å, respectively[41,74–76]. The highly negatively charged nature of the 6HB DNA barrel is compensated in solution by counter-ions such as 12 mM $MgCl_2$ in the cryo-EM analysis.

Cryo-EM imaging of 6HB revealed mono-dispersed particles of the expected nanoscale structure (Fig. 1, Supplementary Fig. 1). The particles were sorted into conformational groups, and the five most populated structures were refined during structural analysis (Fig. 1b, Supplementary Fig. 1). The structures of each group were, on average, resolved at 8 Å, enabling tracing of the six DNA duplex strands in

different conformations, while the most flexible poly-thymidine loops were of lower resolution (Supplementary Fig. 2). All five populations deviated from the ideal, designed 6HB structure and adopted a range of structural conformers. For example, the symmetrical hexagonal arrangement of six duplexes was not found in the structures (Supplementary Fig. 1). This variability in duplex arrangement may be due to the highly flexible duplex interconnection afforded by the terminal crossovers at the top and bottom of 6HB. The multiple and divergent structures reflect that our cryo-EM images (Fig. 1b) are snapshots of largely populated and metastable conformations of the dynamically changing 6HB in solution.

Atomic models for the five 6HB cryo-EM maps were obtained by generating course-grained (CG) molecular dynamics (MD) simulations of the five identified structural classes. The simulations were then flexibly fitted to the cryo-EM maps (Supplementary Table 3) yielding distinct groups termed Class 1–5, with Class 4 being the most populated (see 'Methods'). The overall dimensions for each of the five classes are summarised in Fig. 1c. In general, all classes deviate from the nominal dimensions of the symmetrical 6HB ($90 \times 50$ Å with a 18 Å pore lumen)[42]. Instead, the five classes are shorter and wider with a class average of $67.0 \pm 0.4 \times 71.8 \pm 4.0$ Å, respectively. We relate the widening of the structure to the inherent flexibility of the linkers between the duplexes. The flexibility of the ssDNA poly-T linker regions is apparent by their extended nature and the lack of any secondary structure which allows the duplexes to be as far apart as possible. This separation results in significant gaps between the neighbouring duplexes rather than the regular packing of duplexes within the designed 6HB.

### Conformational changes in DNA helices affect the 6HB architecture

To further elucidate the conformational flexibility of 6HB, we investigated the structure of the six duplexes across the five structural classes. Each DNA helix was analysed independently and aligned with respect to the most populated Class 4 as reference. In this comparative analysis, the six helices adopted the expected B-form DNA structure across all classes (Fig. 2). Nevertheless, the tops of the helices deviated from the ideal structure by duplex fraying which is apparent at helix 5 (Fig. 2). The fraying allows the helices to be further extended away from each other leading to the aforementioned greater overall width of 6HB (Fig. 1c). The structural flexibility of the 6HB helices was also examined by calculating the root-mean-squared deviation (RMSD), relative to Class 4. The comparison across the five classes indicates that helices 2, 3 and, to a lesser extent, 6 exhibit the highest variability (Table 1).

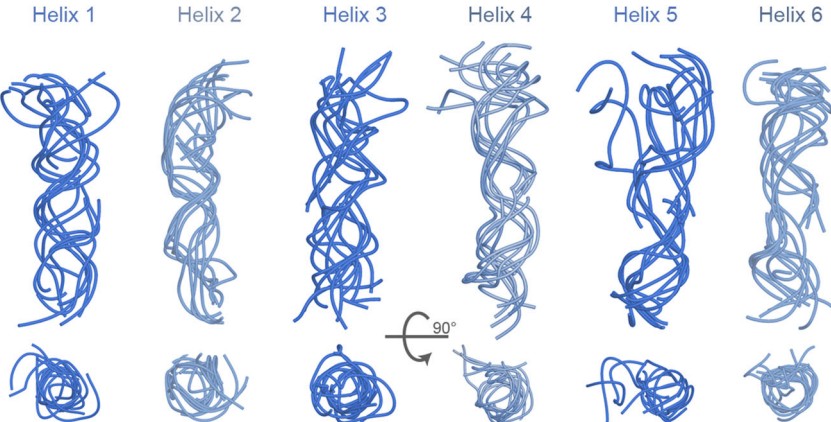

**Fig. 2 | Structural analysis of the six helices of 6HB as obtained from cryo-EM.** The helices from all five classes were aligned using the most populated Class 4 as a reference. All helices are shown in side-on and top-down representations. The helices are colour coded as in Fig. 1.

## The pore lumen is dynamic and variable across different conformations

Following our analysis of the arrangements of the helixes within the porous structures, we investigated the pore lumen. The lumen is functionally important when 6HB is used as a membrane nanopore. Our cryo-EM analysis indicated that the five structural classes feature a bloated cylindrically shaped lumen with a wider centre (Fig. 3). This diverges from the ideal cylindrical channel of the designed symmetric 6HB structure of around 18 Å. The central region of the lumen has an average of 34.0 ± 6.9 Å ranging from 20.7 Å in Class 2 to 40.6 Å in Class 4 (Fig. 3). This represents a span of 19.9 Å. By comparison, the width at the top of the pore ranges from 21.9 Å in Class 2 to 31.0 Å in Class 4, and at the bottom from 19.2 Å in Class 1 to 27.9 Å in Class 3 (see Fig. 3 for averages). This represents a smaller range of 9.1 Å and 8.7 Å, respectively, when compared to 19.9 Å for the flexible centre of the pore lumen. The relative differences in lumen width are related to the nick-region of the component DNA duplex, as explained further below. The cryo-EM-based dimensions of the lumen are wider than the experimentally derived value of 18 Å obtained by size-exclusion with probe molecules[42]. However, the latter value was measured using 6HB inserted inside a lipid bilayer. The structure of 6HB in membranes such as lipid nanodiscs was not explored with experimental techniques in this study.

## Molecular dynamics simulations on the structural variation of 6HB in solution

Our cryo-EM analysis provides a detailed structure for 6HB, yet the insights are based on snapshots of several metastable states of the dynamic architecture. To gain a better understanding of its dynamic nature, 6HB was simulated using CG MD first in solution and later upon insertion into a lipid bilayer membrane. All equilibration and production simulations were performed with the GROMACS 2016 MD engine[95], using the MARTINI force field, version 2 (ref. 96) with the polarisable water model. Further details on the parameterisation of the CG model and other simulation parameters can be found in the 'Methods' and Supplementary Fig. 3. Fifteen trajectories were simulated in a solution of 0.3 M NaCl for a duration of 500 ns apiece, with 5 fs timesteps. In the simulations, temperature and pressure were held at 300 K and 1.01 bar, respectively, with the velocity-rescaling thermostat[97] and the Parrinello-Rahman barostat[98]. To determine if the fifteen parallel simulations were sufficient to generate reproducible results, we confirmed that the standard errors associated with the average pore height and outer width converged with the simulation duration and number of trajectories (see 'Methods', Supplementary Fig. 4). The number and durations of all simulations are provided in Supplementary Table 4.

The simulation results revealed that 6HB is of dynamic nature showing a 'breathing motion' of the duplex bundle (Supplementary Movies 1 and 2). Analysis of the entire ensemble of CG trajectories showed 12 distinct structural clusters. Of these, the 8 most populated were back-mapped to their all-atom (AA) representations. Fitting against the five cryo-EM electron density models resulted in excellent correlation with Pearson correlation coefficients of ~0.8. The good overlap is also apparent from qualitative analysis of the simulated (Fig. 4) and cryo-EM structures (Figs. 1 and 3). This is the case of the structural classes (Figs. 1, 4b) and the averages. For example, the overall shape of 6HB is similar with approximately equal height and outer width of 75.9 ± 0.1 Å and 74.8 ± 0.2 Å, respectively, from MD simulations, and 67.0 ± 0.4 Å and 71.8 ± 4.0 Å from the cryo-EM class average. However, the lumen obtained from simulations (Fig. 4c) resembles a funnel with the widest point at the bottom; the cylinder lumen obtained from cryo-EM was widest at its centre (Fig. 3a). It is worth noting that the dimensions of the lumen obtained by cryo-EM represent the averages across all five classes. Weighting for relative population was not performed due to variations between

## Table 1 | Root-mean-squared deviation (RMSD) of the six duplexes in each class compared to the most populated Class 4

| RMSD compared to Class 4 (Å) | | | | | |
|---|---|---|---|---|---|
| Helix | Class 1 | Class 2 | Class 3 | Class 5 | Average |
| 1 | 3.9 | 4.2 | 6.6 | 5.4 | 5.0 |
| 2 | 4.6 | 13.4 | 13.6 | 13.6 | 11.3 |
| 3 | 13.4 | 13.5 | 12.1 | 6.4 | 11.4 |
| 4 | 7.6 | 4.4 | 4.9 | 3.6 | 5.1 |
| 5 | 14.6 | 4.1 | 4.6 | 3.6 | 6.7 |
| 6 | 6.6 | 3.0 | 12.2 | 12.9 | 8.7 |

Source data are provided as a Source data file.

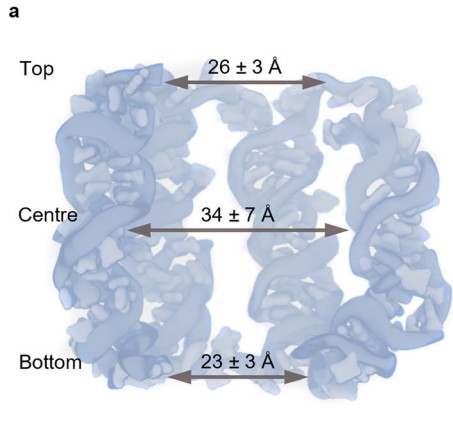

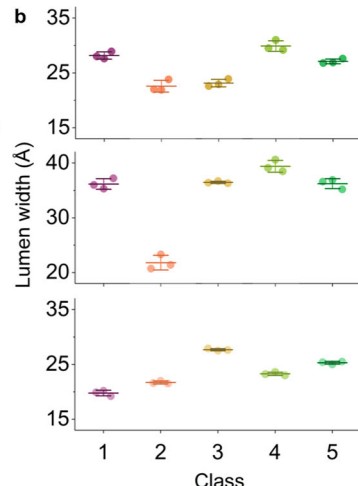

**Fig. 3 | Lumen of 6HB determined from cryo-EM structures. a** Schematic side-on representation of 6HB with two front helices removed. Lumen width was analysed at the top, centre, and bottom, and the average widths (*n* = 3 for each of the 5 cryo-EM classes) are indicated. 'Top' refers to the part of the barrel with longer duplex sections comprising the 5' terminus. **b** Plots showing the average lumen width (with standard deviation) at the top, centre, and bottom of 6HB for each of the five classes (*n* = 3). Source data are provided as a Source data file.

 

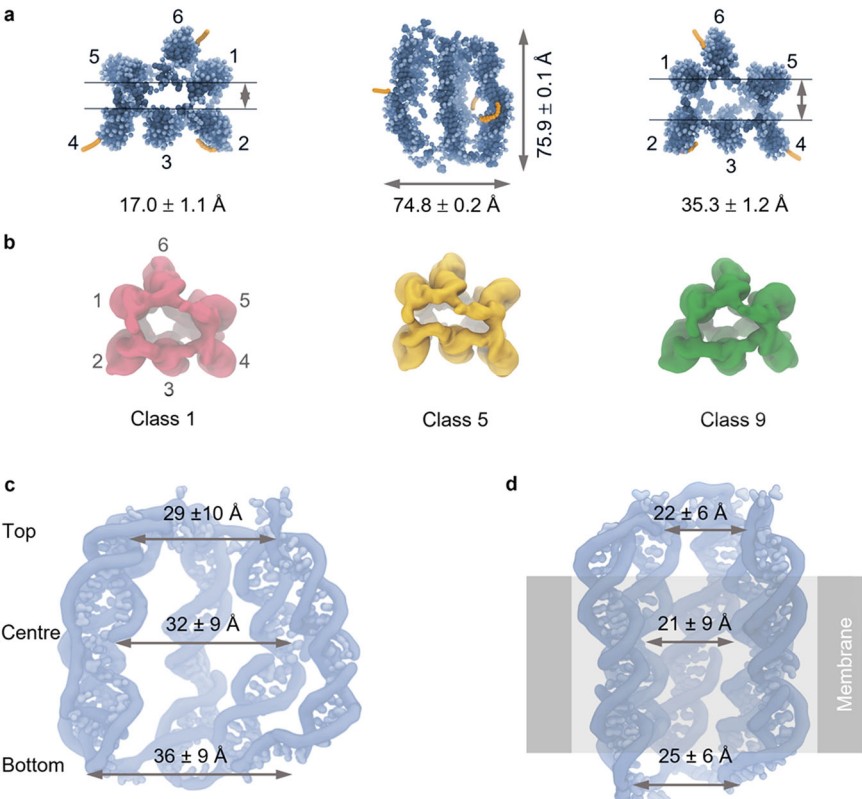

**Fig. 4 | Structural classes and lumen dimensions derived from coarse-grained (CG) molecular dynamics (MD) simulations. a** Averaged CG model of 6HB solvated in 0.3 M NaCl in (left) top-down, (centre) side-on, and (right) bottom-up views. Arrows indicate the minimum width of the pore lumen at the pore opening in the top-down and bottom-up views and the length and width of the pore in the side-on view. The cholesterol anchors are presented in orange. **b** Three of the classes of simulated structures with the best overlap to the cryo-EM structures obtained following all-atom cluster analysis and back-mapping. Class 1 corresponds to cryo-EM class 2, class 5 is close to cryo-EM class 3, and class 9 is similar to cryo-EM classes 4 and 5 with the correlation to 5 higher. All models are shown when viewed from the bottom of the barrel. Numbers indicate the helices. **c** Lumen analysis derived from the CG MD simulations in solution. **d** Lumen dimensions derived from the CG MD simulations of 6HB inserted in a 1-palmitoyl-2-oleoyl-glycero-3-phosphocholine bilayer membrane. The averaged diameter was calculated by taking the mean of the minimum distances between three sets of opposing helices at the top, middle and bottom of the pore. The top, centre and bottom refer to the first, second and third set of 7 base-pairs of the 21 base pair-long duplexes, respectively. Errors are standard deviations as used for the cryo-EM data.

**Table 2 | Helix kink angles calculated from coarse-grain simulation trajectories of the 6HB solvated in 0.3 M NaCl**

| Helix | 1 | 2 | 3 | 4 | 5 | 6 |
|---|---|---|---|---|---|---|
| Helix kink angle (°) | 40.0 ± 0.3 | 62.5 ± 0.5 | 50.8 ± 0.5 | 67.3 ± 0.7 | 62.5 ± 0.8 | 31.0 ± 0.8 |

The averages are standard deviations.

measurements in each class. In contrast, the pore dimension of the simulated structure (Fig. 4a, c) were calculated from the entire ensemble of MD trajectories, implying weighting and statistically more robust results; all following values also represent averages from the entire simulation ensemble.

Combined data from MD and cryo-EM also provide insight into the duplex position and structure. The cryo-EM structures highlighted the flexibility around the strand nick, which resulted in a bloated centre of the pore. The duplexes are also separated from each other by a flexible poly-T linker (Fig. 1a). The MD data, in contrast, highlights that the 6HB structure in solution can also laterally expand (Fig. 4c) relative to the designed elongated symmetrical barrel architecture (Fig. 1a) via the flexible poly-T loops. To further investigate the flexibility of the nick and its affect on duplex structure, we determined the helix kink angles for each of the six helices using our MD models. This analysis indicated that the duplexes are bent by an average of 52.4 ± 5.6° at the single-stranded nick section in the middle of each duplex (Table 2, Supplementary Fig. 5) relative to a straight double-stranded duplex without a nick. Such substantial

bending helps explain the relative flexibility of 6HB in solution as well as why the structure is shorter and wider than the designed elongated barrel.

### MD simulations of 6HB inserted into lipid bilayer membranes
After determining the dynamic structure of 6HB in solution, we explored the effect of lipid bilayer insertion. 6HB insertion can form a functionally important transmembrane channel and is facilitated by cholesterol membrane anchors positioned close to the duplex nicks (Fig. 1a). Our CG simulations in 0.3 M NaCl and a membrane composed of 1-palmitoyl-2-oleoyl-glycero-3-phosphocholine (POPC) reveal that insertion changes the lipid bilayer to form a lipid toroid with its central hole filled by 6HB, in line with reports[91,99,100]. In the toroid, the phospholipids are reorganised causing polar headgroups to make contact with the DNA, thereby minimising unfavourable interactions between the hydrophobic lipid tails and negatively charged backbone. As a consequence of the toroidal arrangement (Supplementary Fig. 6), the upper and lower leaflets of the bilayer become linked to form a contiguous layer[91].

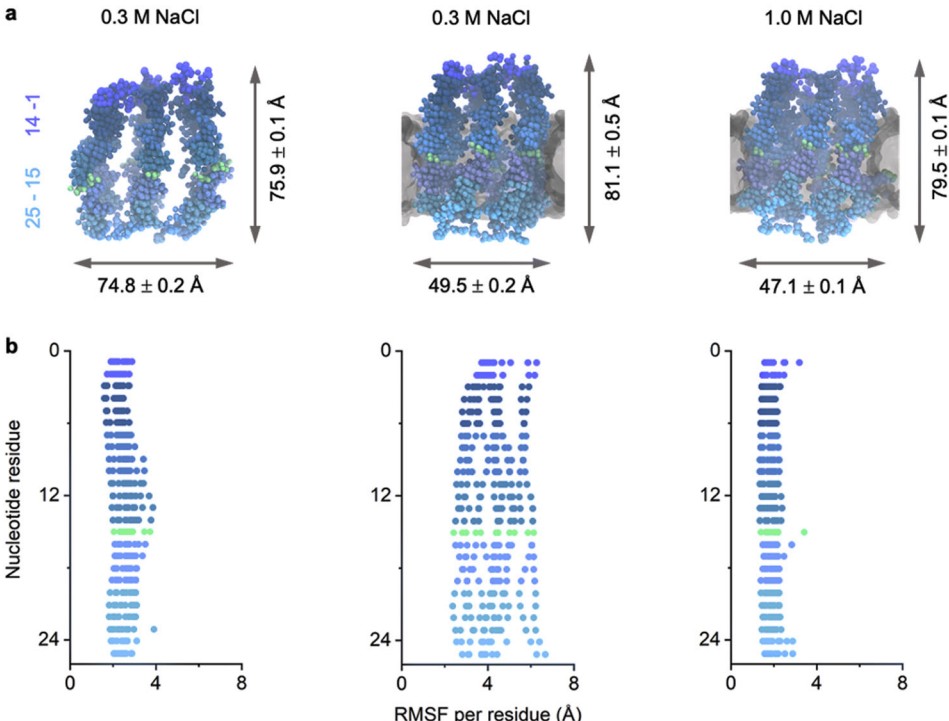

**Fig. 5 | Structure and dynamic properties of 6HB simulated in solution and inserted into a lipid bilayer. a** Averaged structures of 6HB (left) in solution under low salt conditions (0.3 M NaCl), (middle) when inserted into a membrane under low salt conditions and (right) when inserted into the membrane in high salt conditions (1.0 M NaCl). The nucleotide residues 1–14 correspond to the helix portions above the nicks (top); nucleotide residues 15–25 (bottom) are below the nicks. Averages and errors correspond to the standard error of the mean. **b** Corresponding root-mean-square fluctuation (RMSF) per residue plots obtained from simulations using conditions in (**a**). Each dot represents a single nucleotide and is coloured according to its position along the vertical axis of 6HB in (**a**).

Insertion of the 6HB structure into a membrane alters its conformation into a more elongated and tighter barrel that closely resembles the symmetrical 6HB design (Figs. 1a, 4d). The changes are likely caused by lateral membrane pressure. The lateral squeezing alters the 6HB height and outer width to 81.1 ± 0.5 Å and 49.5 ± 0.2 Å, respectively, from 75.9 ± 0.1 Å and 74.8 ± 0.2 Å in solution (Fig. 5a, left and middle). Insertion of 6HB is also associated with other dynamic changes including tilting, twisting, and breathing which are more pronounced in low salt conditions (Supplementary Movie 3 and Supplementary Movie 4). Insertion furthermore remodels the 6HB lumen to become narrower and adopt the expected cylindrical shape (Fig. 4d) by closing of the inter-helix gaps and by straightening of bent duplexes. The effect is most pronounced at the membrane-inserted 6HB midsection where membrane pressure is strongest and the nicked duplexes are most responsive to external influences. The resulting lumen diameter at the midsection of 21.3 ± 0.4 Å (Fig. 4d) is in good agreement with the 18 Å derived by sizing the lumen with probe molecules[42]. The lumen at the top and bottom measures 22.0 ± 0.5 and 24.7 ± 0.5 Å, respectively (Fig. 4d).

**Effect of salt concentration on structural dynamics of 6HB in solution**

After simulating 6HB in a low ionic strength buffer of 0.3 M NaCl, we explored the effect of high salt concentration on the structural dynamics. The chosen salt condition of 1.0 M NaCl is often used in DNA nanotechnology and nanopore applications[41,42,58,73,77]. The CG simulations showed that the average pore dimensions were not strongly affected by the salt concentration (Fig. 5, Supplementary Fig. 7). Similarly, the pore averages of the root-mean-square fluctuation (RMSF)-per-residue were at 39.0 ± 2.5 Å and 42.1 ± 1.3 Å for low and high salt, respectively, within error. Plots of the RSMF values against the residue position (Fig. 5, Supplementary Fig. 7) revealed a more

nuanced picture whereby in high salt the structural fluctuations from nucleotide residues 1 to 14 decreased compared to low salt while those from residues 15 to 25 increased; position 14 is the attachment point for the cholesterol tag. This latter destabilisation is accompanied by cholesterol binding to the hydrophobic groove in the duplex. Similar changes by helix fraying have been reported at higher salt concentrations in previous all-atom (AA) simulation studies of the Drew-Dickerson dodecamer[101].

When considering the region of residues 15–25, the additional measure of percentage of broken base-pairs was increased in high salt conditions (47.3 ± 1.5%) compared to low salt (30.2 ± 3.4%), indicating partial loss of B-duplex character. This marked change is partly a consequence of both the restrained network employed in the MARTINI DNA model which restricts the flexibility of the backbone (Supplementary Fig. 8), and the MARTINI potential parameterisation used for the CG simulations which underestimates the free-energy of DNA base-pairing[96] and hence exaggerates base-pair breakage.

To confirm that the computational setup of CG simulations influences the extent of base pair breakage, all-atom (AA) simulations were conducted (see 'Methods'). In these simulations, the noted restraints of coarse-grain modelling are not present. Indeed, the AA simulations feature a lower percentage of base-pair breakage (Table 3) suggesting that the CG simulations overestimate broken base pairs. Nevertheless, the CG and AA simulations both indicate that helices 1 and 6 are more prone to base-pair breakage (Table 3).

Further analysis of AA simulations on base-pair and base-step parameters explored other structural deviations. For example, helices 1, 4 and 6 exhibited differences to the normal B-DNA values for the stretch, shear and stagger of base-pairs, as well as shift, slide and rise of base-steps (Supplementary Table 5, Supplementary Figs. 9–11). These deviations were most pronounced in the base-pairs and steps in immediate vicinity of helix nicks and crossovers, indicating their

**Table 3 | Percentage of base-pair breakage (BPB) within the coarse grain (CG) and all-atom (AA) simulation ensembles of 6HB dissolved in 0.3 M NaCl**

|  | CG % BPB | AA % BPB |
|---|---|---|
| Residues 1–14 | 45.7 ± 2.1 | 15.0 ± 1.1 |
| Residues 15–25 | 30.2 ± 3.4 | 14.4 ± 1.5 |
| Helix 1 | 60.4 ± 3.0 | 18.9 ± 1.3 |
| Helix 2 | 5.58 ± 1.66 | 10.8 ± 1.6 |
| Helix 3 | 67.8 ± 1.8 | 10.4 ± 3.1 |
| Helix 4 | 16.4 ± 1.0 | 14.6 ± 0.7 |
| Helix 5 | 44.0 ± 3.3 | 12.8 ± 1.2 |
| Helix 6 | 53.6 ± 3.4 | 20.0 ± 2.6 |

Errors represent the standard error of the mean.

contribution to the destabilisation of the 6HB structure. In support, DNA helix instability at the strand nicks with a cholesterol modification has been reported previously[102].

After revealing fluctuations of individual bases within a duplex, we also assessed the dynamics of each duplex in CG simulations by the per-duplex RMSF. The results show that unmodified duplexes 1, 3 and 5 had less structural variation than the cholesterol-modified duplexes 2, 4, and 6 (Supplementary Fig. 12). This difference was more pronounced in low salt conditions. The higher fluctuation of the cholesterol-modified duplexes is likely due to the interaction of cholesterol with base-pairs in the frayed nick region (Supplementary Fig. 13).

### Effect of salt concentration on structural dynamics and stability of membrane-inserted 6HB

The effect on salt concentration was also examined for the membrane-inserted helix bundle. When compared to low salt conditions, the dimensions of membrane-inserted 6HB in high salt were comparable with $79.5 \pm 0.1\,\text{Å} \times 48.1 \pm 0.1\,\text{Å}$ (Fig. 5a, right). The simulations in the two conditions differed, however, as 6HB was eventually ejected from the membrane in two-thirds of the trajectories in low salt conditions. In these simulations, 6HB transitioned from a transmembrane state, where all three cholesterol lipid anchors interact with the membrane (Supplementary Fig. 14, 100 ns) to a membrane-tethered state mediated by one or two cholesterol anchors (Supplementary Fig. 14, 800 ns). Membrane expulsion was preceded by lipid reorganisation and pore twisting, tilting, and breathing (Supplementary Movie 5). By contrast, 6HB remained membrane-inserted in all high salt simulations. These observations suggest that 6HB membrane insertion is energetically unfavourable in low salt conditions, something which has so far not been detected in experiments[75,76]. Observing membrane expulsion events during 300–800 ns was only possible in CG simulations as running all-atom simulations at these long timescales would be computationally prohibitively costly.

To understand why membrane insertion is less favourable in the low salt condition, we investigated the per-residue RMSF of bilayer spanning 6HB. In these simulations, the entire DNA backbone fluctuated considerably, much more than in solution (Fig. 5b, middle). By contrast, the RMSF in high salt conditions was strongly reduced when compared to membrane-inserted 6HB in low salt conditions (Fig. 5b, right). The observed link between membrane expulsion and duplex fluctuation is related to the compression of the inserted 6HB duplex bundle by the surrounding lipid-dense toroid[91,103]. Compressing the negatively charged duplexes into close proximity is energetically unfavourable when backbone charges are not sufficiently electrostatically screened in low salt conditions, as indicated by the high RMSF. In response to this sterically constrained inter-duplex repulsion, the inserted 6HB is expelled from the membrane. In contrast, enhanced electrostatic screening at high salt conditions

alleviates the repulsion between DNA helices, allowing them to come closer together. This is evident from the lower RMSF and the drastic reduction in the average inter-helix distance of ~9 Å upon increasing the salt concentration (Fig. 5b, right). As further observation, the fluctuations of 6HB in the CG simulations are dampened compared to those of in the AA simulations (Supplementary Fig. 8) due to the "stiff" elastic network of restraints employed in the MARTINI model of DNA[96].

After determining how salt concentration alters the 6HB structure, we explored the distribution of salt within the water-filled pore lumen relative to bulk solvent. The simulation results depict the membrane and the sodium and chloride ions within the simulated box-shaped boundaries (Fig. 6a, b) but lack the DNA nanopore for reasons of visual clarity. The analysis for high and low salt conditions reveals a higher number density of sodium ions within the membrane torus where the 6HB pore resides (Fig. 6a–c, Supplementary Table 6) compared to the buffer solution outside the torus. Apparently, the Na$^+$ ions are concentrated by the negatively charged DNA pore (Fig. 6a–c). The increase in sodium ion density from outside the torus to the 6HB lumen is more pronounced in high salt ($+0.19\,\text{nm}^{-3}$) than in low salt ($+0.15\,\text{nm}^{-3}$) (Fig. 6c). Similarly, water concentrates within the membrane torus (Fig. 6c). Two additional water density peaks on either side of the central peak correspond to the perimeter between the outside of the pore and the surface of the membrane torus (Fig. 6c). The increase in water density towards the lumen of the membrane torus is lower in high salt ($1.8\,\text{nm}^{-3}$) than in low salt ($2.3\,\text{nm}^{-3}$) (Fig. 6c) as the higher density of ions causes water to be excluded from the pore lumen. In both salt conditions, the number density of chloride ions remains relatively uniform, albeit with a slight depletion in the proximity of the nanopore in 0.3 M NaCl (Fig. 6a–c) due to the repulsion by the negatively charged DNA pore.

Further analysis confirmed that our course-grained simulations using the MARTINI model with polarisable water provide a reliable description of local counterion densities in the proximity of DNA duplexes[95,104]. In the MARTINI force field, sodium ions tend to aggregate equally along the backbone and within the grooves of the DNA, whereas in the all-atom CHARMM force field the ions exhibit a stronger preference for the grooves[96]. Notwithstanding this feature, the radial distribution functions between MARTINI cations and highly charged polymers reproduce their AA equivalents well.

### Computational electrophysiological analysis of membrane-inserted 6HB

Following our analysis of the lumen profile and pore dynamics of membrane-embedded 6HB, we investigated the electrophysical channel properties. Current recordings are a common means of assessing channel properties experimentally[41–43,47,77,90]. To compare simulation and experiment, we calculated the current passing through our ensemble simulations at voltages ranging from 0–100 mV using the computational electrophysiology protocol implemented in GROMACS[105,106] (see 'Methods'). Using ensemble methods and the associated statistical analysis provides error bars by which the precision of these calculations can be assessed. While a systematic study of the sources of uncertainty in the model is beyond the scope of this work[107–109], it is anticipated that the lumen diameter and properties affecting the hydrophobicity of the DNA pore are among the parameters that dominate the uncertainty in our simulations.

The current–voltage curve (Fig. 7a) from the computational electrophysiology yielded a conductance of $1.19 \pm 0.49$ nS, which is within error of the experimental conductance of $1.61 \pm 0.09$ nS[42]. The data points from computational electrophysiology are scattered more widely than in experiments which is a consequence of the stochastic nature of the simulations. The histogram for simulated conductances (Fig. 7b) exhibited a higher and a lower sub-conductance state (1.5 nS and 0.5 nS) as found in experiments[42]. The higher conductance reflects a pore with six fully formed duplexes while the lower conductance is

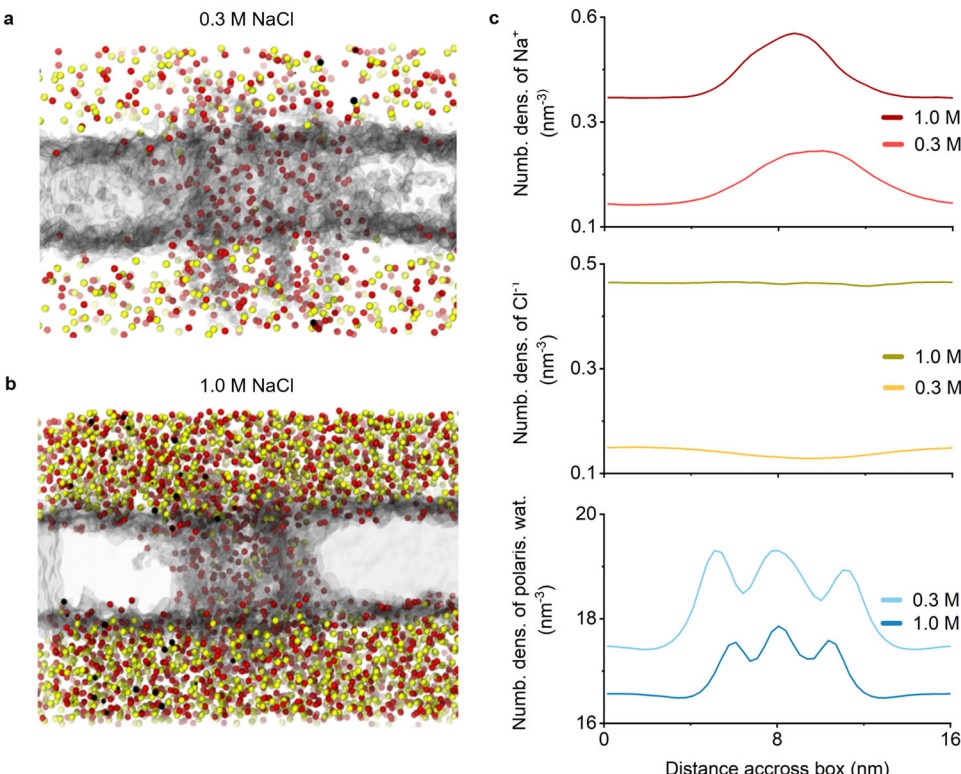

**Fig. 6 | The distribution of water and buffer ions in the membrane-spanning 6HB lumen in low and high salt conditions. a, b** Snapshots taken after 1 μs of simulation time in **a** 0.3 M NaCl and **b** 1.0 NaCl. Red spheres represent the sodium ions and yellow spheres represent chloride ions. The simulation images are 16 nm wide. **c** Number density plots of sodium ions (top), chloride ions (middle), and MARTINI polarisable water molecules (bottom) along the vertical axis across the simulation box, calculated for the first 300 ns of the simulation ensembles for membrane-spanning 6HB. Data after 300 ns were not used as the nanopore typically begins to depart from the bilayer in two-thirds of the 0.3 M NaCl trajectories. Source data are provided as a Source data file.

caused by structural rearrangement of 6HB at high voltage. Under such conditions, the high electric field electrophoretically moves parts of the negatively charged DNA barrel out of place. The good agreement with experimental conductance histogram data further validates the simulations and also extends previous simulations that reveal the current–voltage relationship of protein pores and DNA nanopores[110].

## Discussion

Our study has combined single-particle cryo-EM analysis and modelling simulations to understand the structural dynamics of an archetypal six-helix bundle DNA nanostructure. Our analysis established good consistency between the two approaches. Cryo-EM showed that the six-helix bundle has a dynamic structure where DNA duplexes are flexible within the bundle without adopting the rationally designed hexagonal arrangement. Coarse-grained molecular dynamics simulations were in good agreement with the cryo-EM results. The duplex bundle experiences duplex fraying at duplex interlinks as well as bending resulting in a more compact structure than the elongatedly designed cylindrical barrel. The dynamic analysis furthermore exhibits the dynamic, breathing motion of the barrel. Insertion of 6HB into the anisotropic environment of the bilayer membrane, however, drastically shifts the bundle structure into the hexagonal and parallel aligned DNA duplexes of the design. The simulations also demonstrate a significant effect of salt concentration on the stability of the small 6HB when inserted into a lipid bilayer membrane. Future studies may explore the DNA nanostructure inserted into lipid nanodiscs with cryo-EM. In conclusion, our dynamic structural insights will be critical in the rational design of DNA nanostructures for the development of targeted applications for research, technology, and medicine. The benefit of combining simulations and cryo-EM in order to maximise the value of structural investigations may also be extended to analyse other DNA nanostructures.

## Methods

### Cryo-EM

**Sample preparation and data collection.** The 6HB DNA nanostructure (3 μL at 1 μM in 12 mM $MgCl_2$, 0.6 x TAE, pH 7.4 buffer) was applied to C-flat (2/2 μm) holey carbon grids (EM Sciences, Hatfield, PA, USA). Grids were incubated for 20 s at 10 °C and 100% humidity in a Vitrobot Mark IV (ThermoFisher, Milton, UK), blotted for 6 s before plunge-freezing in liquid ethane for vitrification. Cryo-EM grids were then clipped and transferred to an auto-loader for data collection. Data for 6HB was collected on a 300 kV Krios G3i microscope (FEI/Thermo Fisher) with a post-GIF (20 eV slit), using EPU software v.3.0 (ThermoFisher Milton, UK), and equipped with K2 Summit direct electron detector (Gatan Inc., Pleasanton, CA, USA) at the Birkbeck EM facility (ISMB, London). Movies (50 frames per movie) were collected with a dose of 1.2 e⁻/Å² per frame with a calibrated pixel size of 1.05 Å/pixel. Images were collected at a range of defoci between −1.2 to −2.5 μm. In total, 5567 movies were collected from a single cryo-EM grid.

**Data processing.** All movies were imported into RELION 3.0 (ref. [111]) and aligned using MotionCorr2 (ref. [112]). CTFFIND4 (ref. [113]) was used to determine defocus values. The resultant micrographs were inspected and screened for CTF quality if any aberration effects or unreliable CTF estimation were present. In total, 4520 micrographs were selected for further processing. Particle picking (ppick) was done using crYOLO v1.3.6 (ref. [114]): a set of 50 randomly selected micrographs was used for the initial manual picking of particles, coordinates of which were imported into the crYOLO training model (based on the general training network (ref. [114]) allowing to optimise automatic ppick

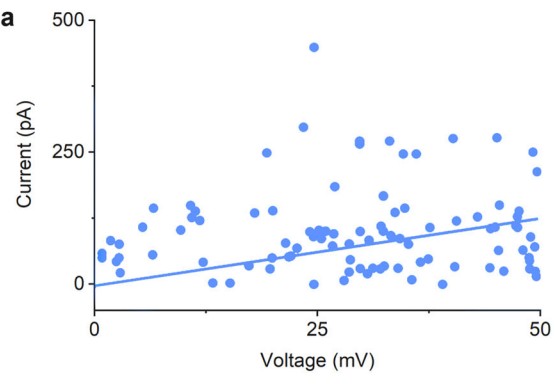
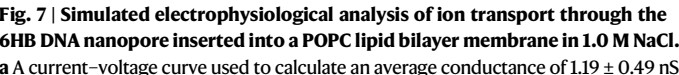
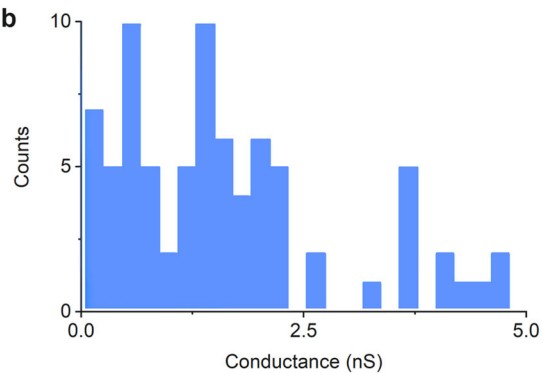

**Fig. 7 | Simulated electrophysiological analysis of ion transport through the 6HB DNA nanopore inserted into a POPC lipid bilayer membrane in 1.0 M NaCl.** **a** A current–voltage curve used to calculate an average conductance of 1.19 ± 0.49 nS

(standard error of the mean). Data points from simulations are shown along with the fitted trend lend. **b** Conductance distribution plot illustrating the distribution of conductance states.

parameters. For this procedure, micrographs were low-pass filtered to 20 Å. The trained ppick parameters (threshold value 0.35) were optimised by running several iterations and subsequently testing the quality of the ppick with the new parameters on a sub-set of 100 randomly selected micrographs. The optimised ppick parameters were used for processing the entire dataset by crYOLO. The picked coordinates of particles were exported into RELION (ref. 111) for extracting the particle images (PI) from the micrographs using a box size of 300 × 300 pixels. In total, ~368,501 PIs were extracted. These PIs were exported into cryoSPARC v2.9.0 (ref. 115) and subjected to two-dimensional (2D) classification. All subsequent steps in image processing were carried out in cryoSPARC, unless stated otherwise. Several rounds of two-dimensional classification were performed to select images containing only particles with structural features characteristic to 6HB; 2D classes were showing different orientations (Supplementary Fig. 1). Images with no features reflecting the 6HB complex were discarded. The selected particle images corresponding to 6HB in different orientations were subjected to ab-initio 3D classification in cryoSPARC in two rounds, separating the images in ten initial 3D maps (using K = 10 seeds) in each round. A resolution of the best initial map was 15 Å. Then the first round of refinement was done with respect to this model and followed by the first 3D classification. Five maps revealed features characteristic to the 6HB duplex, and particle images used to calculate these structures were extracted and grouped for the following processing (Supplementary Fig. 1). These selected particle images were subjected to 3D analysis using an option "homogenous" 3D refinement, implemented in cryoSPARC. The second round of 3D classification exposed five distinct conformations of the 6HB complex. To further improve the alignment of particles and attain improved 3D maps 'Local 3D refinement' was performed, using masks obtained from the initial homogenous refinement. This allowed us to improve the resolution in the refined cryoEM maps up to 7–8 Å at 0.143 Fourier-Shell correlation (FSC) threshold criterion based on a gold-standard approach (ref. 116). The number of particles in the classes was Class 1: 16,938 particles, Class 2: 25,871 particles, Class 3: 24,318 particles, Class 4: 27,900 particles, Class 5: 22,799 particles. Local resolution of each class was assessed using ResMap (implemented in RELION (ref. 117) (Supplementary Fig. 2). Both the non-sharpened and sharpened maps (from cryoSPARC) were used for model fitting and refinement (Supplementary Table 3, spreadsheet 'cryoEM struct.refin.statist.' in Source data file).

**Model building and validation.** Fitting into the individual 6HB 3D maps was done using molecular dynamics (MD) simulation coarse-grained models as follows: Ten atomic models obtained from MD simulations under different salt conditions and clustered according to conformations were used for rigid-body fitting in the five cryo-EM

maps, using UCSF CHIMERA v1.14 (ref. 118) 'Fit in Map' tool. The fitting of each model into the map was assessed based on the cross-correlation score between the map and the models, at a resolution 8–10 Å (Supplementary Table 3). In total, three MD-simulated models showed the highest cross-correlation scores for model-map fittings. The models with the highest cross-correlation scores with the maps were selected for each cryo-EM map for further flexible fitting and refinement. In order to improve the correspondence of the six DNA helices with densities in the cryo-EM maps, we used firstly normal-mode analysis in iMODFIT v.1.44 (ref. 119) and subsequently ISOLDE v1.3 (ref. 120). In the final step, models were subjected to real-space refinement using PHENIX v1.14 to fine-tune the model fittings[121]. Segments of single-stranded DNA that connect the inter-linking DNA strands were less accurately fitted due to the limitations in the cryoEM map resolution. The quality of the fittings was inspected manually using COOT v0.8.9.1 (ref. 122) and later through model validation tools using PHENIX v1.14 (ref. 121). All figures were produced using UCSF CHIMERA v1.14 and CHIMERAX v1 (ref. 123).

### Molecular dynamics simulations

**All-atom modelling of 6HB.** To gain a better understanding of its dynamic nature, 6HB was simulated using coarse-grained (CG) molecular dynamics (MD) first in solution and later upon insertion into a lipid bilayer membrane, using an all-atom model of 6B as starting point. Fifteen trajectories were simulated to improve the statistical validity of the simulations. Before any CG models were built, an all-atom (AA) model of 6HB was constructed and the covalent linkages between the pore and the tri(ethylene glycol) linked cholesterol (TEG-C) were parameterised. Helices 1–6 were built with the Nucleic Acid Builder module in AMBERTools16 (ref. 124) and arranged hexagonally with a ~2 nm inter-helix spacing using PyMOL version 1.2r3pre (ref. 125). The inter-helix crossovers, TEG-C anchors and the covalent linkages between TEG-C and 6HB were also made using PyMOL utilities. The TEG-C anchors and the covalent linkages to 6HB were parameterised within the CHARMM General Force Field (CGenFF)[126]. NAMD v2.12 (ref. 127) was used for all AA simulations, in conjunction with the CHARMM36 and CGenFF 3.0.1 force fields. 6HB was solvated in a 35 × 35 × 40 nm box of TIP3P water molecules using the VMD 1.9.3 solvate plugin[128], and the autoionize plugin was used to set the concentration of KCl to 0.3 M. The system was minimised and then equilibrated for 25 ns with harmonic restraints to slowly relax the structure. The force constant (k) of these restraints was gradually reduced throughout the equilibration, from 0.5 kcal/mol/Å² to zero. The model was then subjected to another 50 ns of unrestrained dynamics. The temperature was set to 300 K with the Langevin thermostat, and isotropic pressure control at 1.013 bar was achieved using the Nosé-Hoover Langevin piston method. Electrostatic forces were

calculated using the particle-mesh Ewald (PME) algorithm, and a 1.2 nm cutoff was used for vdW and electrostatic interactions.

**Coarse-grained modelling of 6HB in solution.** A relaxed barrel-like configuration of 6HB was generated from the first 25 ns of the AA equilibration trajectory, and this structure was converted to its MARTINI[66,96] coarse-grained representation using the martinize-DNA.py (version 2.6) script[96]. The script automatically generates numerous elastic restraints optimised to maintain the base-pairing, general structure and persistence length of double-stranded B-DNA. The 'stiff' elastic network was used for all CG simulations which applies flexible restraints to all of the pseudo-atom beads in order to reinforce base-pairing and base-stacking was used in these simulations. The MARTINI model allows non-parameterised system components such as the TEG-C anchors to be built by mapping biomolecular all-atom structures to their coarse-grained representation, typically by grouping 3–4 heavy atoms into one MARTINI "bead" and defining the connectivity between the beads. MARTINI beads are classified according to four types: apolar, polar, non-polar and charged. Each bead type is associated with a characteristic set of non-bonded (i.e. Coulomb and Lennard-Jones) parameters. Therefore, bead types with their associated non-bonded parameters must be custom-assigned according to chemical characteristics. Bonded parameters are extracted from reference AA simulations, making use of the Boltzmann inversion method to derive coarse-grained bonded potentials. This procedure provides a coarser representation of the distribution of heavy atom bond lengths, angles and dihedrals than would be observed in a fully atomistic model.

An atom-to-bead mapping was constructed for the TEG-C anchors (Supplementary Fig. 3). MARTINI bonded parameters for the TEG-C anchors were generated with the PyCGTool 1.0.0 program developed by Graham et al.[129]. Radius of gyration ($R_g$) data for the CG TEG-C anchors were generated from a set of five unrestrained CG simulation of 6HB in solution, was compared to the ($R_g$) data taken from a set of five 30 ns AA reference simulation of the NP in solution. For each anchor, the percentage difference between the total $R_g$ from the AA reference simulation and the total $R_g$ from the CG simulation was under 5%, indicating a good agreement between the results yielded with the CG parameters and those with AA parameters.

All CG simulations were performed with GROMACS v5.1.4 (refs. 95,130). Temperature control at 300 K was achieved with the use of the velocity-rescale thermostat with a time constant of 1 ps for equilibration and production. Two different barostats were employed for pressure control (at 1.013 bar) during NPT simulation phases. The Berendsen barostat[131] was used for early equilibration (with time constant of 2.0 ps) to relax the volume and pressure of the simulation box. As the Berendsen barostat does not allow the box shape to change, it does not yield a true NPT ensemble Hence, the Parrinello-Rahman barostat[98] was used for production/data collection. Replica production simulations, as required for ensemble-based protocols (see below) were run by generating separate run files with different random velocity seeds. Velocities, energies, forces and coordinates were recorded every 40 ps, corresponding to 25 'snapshots' per ns. All CG simulations employed the leap-frog integration algorithm, and the reaction-field method was used for calculating electrostatics. A 1.1 nm cutoff used for vdW and short-range electrostatics, with the neighbour-list being updated every 20 steps.

**Coarse-grained modelling of membrane-spanning 6HB.** The solvated membrane models were built by inserting CG TEG-C-modified 6HB into the centre of a pre-equilibrated CG POPC bilayer patch, which was solvated in a box of polarised MARTINI water molecules with the insane.py script[132]. CG Na$^+$ and Cl$^-$ ions were added to set NaCl concentrations of 0.3 M and 1.0 M NaCl, also with the insane.py script. A POPC bilayer was selected as they are used extensively in electrophysiology experiments, and POPC lipids make up a large proportion

of the lipids found in eukaryotic cell membranes. The final models consisted of ~76,000 pseudo-atoms, with overall cell dimensions of $16 \times 16 \times 18$ nm.

The initial models were subjected to 5000 steps of minimisation using the steepest-descent algorithm. A 2 fs time-step was used for the first 10 ns of equilibration time, as the initial relaxation of system involves drastic conformational changes which result in large forces being calculated at each time-step. For the NVT phase of the equilibration, strong positional restraints were applied to the beads of 6HB and the POPC head-groups. The force constants of TEG-C-modified 6HB restraints were gradually reduced to zero over the course of 4 ns. The temperature was held at 300 K throughout the equilibration. The next phase of equilibration was performed in the NPT ensemble with the Berendsen barostat, to relax the volume of the system while slowly releasing the restraints on the POPC head-groups, to prevent the formation of voids within the bilayer. This was done over 6 ns of simulation time, after which all positional restraints were removed. The time-step was increased to 5 fs, and the unrestrained system was subjected to another 20 ns of simulation time with the Berendsen barostat. For the final phase of equilibration (lasting 70 ns) the Parrinello-Rahman barostat was employed with a time coupling constant of 12 ps. Production simulations were then run for 1 μs, with temperature and pressure held at 300 K and 1.013 bar, respectively.

**Reproducibility and uncertainty quantification.** Individual MD trajectories are highly sensitive to their starting conditions, and the temporal evolution of an individual trajectory is stochastic, meaning that neighbouring trajectories with different initial velocities diverge quickly. One cannot assume that the conformational space has been sufficiently sampled from a handful of repeat simulations without performing appropriate error analyses on the computed data to determine whether or not the computed average is representative of the true ensemble average. Therefore, the reliability of the results produced in such studies is uncertain. We address this issue by employing an ensemble-based protocol, according to which a set of N "replicas" are run concurrently, producing a stable ensemble average and associated fluctuations such that running N+1 would not alter the behaviour significantly. Running ensembles of computationally inexpensive CG simulations enhances the sampling of conformational space, giving reliable results in a rapid and reproducible way. For uncertainty quantification, we have employed the bootstrap method for calculating the standard error associated with ensemble-averaged macroscopic properties of the CG 6HB systems; namely the pore height (for all four models), and the bilayer thickness (for the membrane models). Ensemble average plots illustrating this analysis are shown in Supplementary Fig. 4. This statistical method has been utilised successfully in many ensemble-based all-atom MD studies[67,133–137], where the typical simulation duration is fairly short (4–20 ns), but its use in CG studies at longer timescales has not previously been reported. In this work, we have achieved similar success to previous AA studies in terms of error control and reproducibility.

**Computational electrophysiology simulations.** A double-bilayer system was constructed by cloning the fully equilibrated 1.0 M NaCl 6HB/POPC system in the z-direction, such that two distinct solvent compartments (A and B) were formed, corresponding to the *cis* and *trans* compartments in a traditional chip-based parallel bilayer recording instrument. Virtual cylinders were defined around the two parallel membrane-spanning pores, a point charge difference (Δq) was applied to the system in each simulation, and the net influx of cations and anions into each compartment was recorded every 0.1 ns. Whenever the ion count in each compartment differed from the reference count (defined by Δq), ions from one compartment were swapped with water molecules from the other compartment to restore the reference count. Each simulation was run for 100 ns, and each trajectory was split into 20 ns

slices[105,106]. For each trajectory slice, the potential difference (V) was calculated using the gmx potential[95] tool, and the instantaneous current flowing through each pore (I) was calculated according to the following expression (Eq. 1):

$$I = \frac{\Sigma_i (q_i \times n_i \text{ of swaps})}{2 \triangle t} \qquad (1)$$

where $q_i$ is the charge of the ion, multiplied by the number of ion/water swaps for that ion. Thirty replicas were performed in total; one ensemble of ten simulation replicas with $\Delta q = 0$, an ensemble of ten with $\Delta q = 2$, and another ensemble of ten with $\Delta q = 4$. The frequency of ion/water swaps caused the instantaneous potential difference to fluctuate sharply throughout each simulation, so some of the trajectory slices were omitted as the potential difference rose beyond the range of voltages that are typically employed in electrophysiology experiments on DNA nanopores (±100 mV). To improve the comparison of the electrophysiology simulation with experiments, the voltage conditions in simulations are consistent with those used in experiment. Hence, only trajectory slices within a voltage range from −100 mV to +100 mV were considered. Around 100 usable data points were extracted from the full set of trajectories, and these were invoked to construct a current–voltage (IV) curve. Bootstrap error analysis confirmed that the usable trajectory slices derived from these thirty replicas yielded a converged value for the average conductance G, which was taken as the slope of the line fitted to the IV curve.

**Analysis of base-pair and base-step parameters from AA simulations.** Averaged parameters were calculated for each helix from the ensemble of 15 × 30 ns AA simulations of 6HB in 0.3 M NaCl using the x3DNA DSSR software[138,139]. The base-pair parameters showed some deviation from the idealised values for B-DNA due to base-pair breakage that occurs near the nick sites and inter-helix crossovers (Supplementary Fig. 12). The averaged values for the tilt, roll, shift, and slide parameters for each of the helices were in good alignment with previously published base-step parameters calculated for a series of nicked DNA 42-mers[140] and various paranemic crossover DNA nanostructures[141], however for helix 4 and 6 the twist and rise parameters, which stabilise the helical structure of DNA[141] are particularly aberrant, indicating substantial loss of B-DNA structure in these helices (Supplementary Fig. 13).

**Reporting summary**
Further information on research design is available in the Nature Portfolio Reporting Summary linked to this article.

## Data availability

The data that support this study are available from the corresponding authors upon request. Cryo-EM maps are available from the Electron Microscopy Data Bank (EMBD) under accession codes EMD-14342 (Class 1), EMD-14343 (Class 2), EMD-14344 (Class 3), EMD-14345 (Class 4), EMD-14346 (Class 5). Atomic models have been submitted to the Protein Data Bank (PDB) under accession codes 7YHW (Class 1), 7YWI (Class 2), 7YWL (Class 3), 7YWN (Class 4), 7YWO (Class 5). The source data underlying Table 1, Fig. 3, and Fig. 6 are provided as a Source data file. Source data are provided with this paper.

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

## Acknowledgements

This research was supported by the BBSRC (BB/M012700/1 and BB/M025373/1 to S.H., BB/M009513/1 to C.L.). K.A. was funded by an EPSRC (DTP) PhD studentship. The simulation work led by P.V.C. was performed on ARCHER, the UK's national High Performance Computing service, and Cartesius, hosted by the Dutch national High Performance Computing centre SURF. The funding for the computational work was provided by CompBioMed and CompBioMed 2 (http://www.compbiomed.eu, Grant Nos 675451 and 823712), and the UCL Provost. EM analysis led by E.O. was performed using the ISMB EM facility at Birkbeck College (University of London) funded by the Wellcome Trust (202679/Z/16/Z and 206166/Z/17/Z). We acknowledge Dr. Natalia Lukoyanova with Dr. Shu Chen (Birkbeck) for their help with the data collection for this investigation at the ISMB EM facility at Birkbeck College. We thank Dr. David Houldershaw and Yanni Goudetsidis for computer support at Birkbeck throughout the duration of the project, Dr. Jane Denyer for providing useful comments on the manuscript, and Dr. Jonah Ciccone for helping prepare the figure illustrations.

## Author contributions

The study was designed by E.O., P.V.C. and S.H. The DNA structures were prepared by C.L. A.J. performed cryo-EM analysis and K.A. conducted the simulations. The figures were prepared by A.J., K.A. and C.L. The paper was written by C.L. and S.H. with input from the other authors.

## Competing interests

The authors declare no competing interests.
