## [Peer Review File · Nature Communications]

Structure and dynamics of an archetypal DNA nanoarchitecture revealed via cryo-EM and molecular dynamics simulationsReviewers' Comments:

Reviewer #1:

Remarks to the Author:

In this study, Ahmad et al. combine cryoEM imaging and coarse-grained molecular dynamics simulations to characterize in great detail the structure of a six-helix DNA bundle (6HB) in solution. Molecular dynamics simulations are then used to explore the structural consequences of the insertion of the bundle into a lipid bilayer membrane at different salt concentrations. Insertion into the lipid bilayer is mediated by cholesterol membrane anchors positioned close to the duplex nicks.

The results show that in solution the 6HB is rather flexible and assumes conformations that deviate from those expected from design, being typically wider and shorter and with a larger funnel-shaped lumen. Contrarily, insertion of the 6HB into the lipid bilayer results into the reconfiguration of the bilayer into a toroid that laterally squeezes the 6HB, making this adopting a structure that is much closer to the theoretical expected from design. Insertion into the lipid bilayer is also more stable at high salt concentrations.

This is a well-done and accurate simulation study on a relatively simple DNA structure, which essentially confirms and explains the experimental data available from the literature. I do not have special remarks besides a few comments below:

p. 5: „Our cryo-EM analysis indicated that all five classes feature a funnel-shaped lumen (Figure 3) which is different to the cylindrical lumen of the original 6HB design”
I would recall here again the dimension expected for the lumen (18 Å).

“The funnel is narrowest at the top of the 6HB with a diameter of 42 ± 5 Å and narrowest at the bottom with a width of 26 ± 2 Å.” This sentence should be probably reformulated as: “The funnel is narrowest at the top of the 6HB with a diameter of 26 ± 2 Å and widest at the bottom with a width of 42 ± 5 Å”

Figure 3B: the y-axis, should be maybe labelled as “Lumen Width” instead of “Lumen Length”?

P6: “...6HB is wider than the indirectly determined lumen width of 18 Å”. I would replace with “...6HB is wider than the nominal value of lumen width, which is 18 Å.”

Reviewer #2:

Remarks to the Author:

The manuscript from the Howorka group uses cryo-EM combined with coarse-grained molecular dynamics simulations to investigate the structure of a DNA nano-architecture embedded in a membrane. CryoEM studies have revealed a wealth of new information for DNA nanotechnological structures which have been of high scientific impact, as recently reviewed by Pan et al in Small (<https://doi.org/10.1002/sstr.202100053>). My expertise is in computer simulations of proteins and nucleic acids, so I cannot comment on the appropriateness of the experimental methodologies. It is extremely nice to see that computer simulation is playing an increasingly important role in complementing cryo-EM studies, as has been done in the work here. Clearly, the two techniques provide synergistic information that increases insight and understanding.

There are some aspects of the descriptions of the simulation methodology that it would be helpful to clarify. I found it difficult to understand from reading this version of the paper which simulations (atomistic/coarse-grained/in solution/in bilayer) were performed and why. It might be helpful to include a table summarising this in the Methods or Supplementary Information.

I was surprised that most of the structural and dynamic analysis reported focused on the coarse-grained simulations and did not include data from the atomistic models. This would be of interest as both length-scales provide complementary information, although more than 50ns of atomistic data would be required (which should not be computationally unfeasible using modern GPU resources). Having visually inspected the pdb files there was a striking lack of complementary base pair hydrogen bonding and base stacking. The authors should provide a quantitative description of the helical parameters in the atomistic DNA structures, e.g. using x3DNA or Curves++, to characterise the base-pairing, stacking and backbone conformations in comparison to canonical B-form DNA, and other DNA nanotechnology structures in the literature. If there are significant deviations, then the authors should explain what key inter-atomic interactions are stabilising their structures.

From my reading of the paper, the level of restraints imposed on the coarse-grained DNA simulations as reported in the Supplementary Information is ambiguous. If restraints are indeed needed to maintain the duplex, then how does this affect the dynamics and fluctuations of the DNA reported in Figure 5, and elsewhere? Please could the authors clarify this, and explain any potential artifacts that may have been introduced.

The results showing that the DNA-nanostructure is unstable in the lipid bilayer under low salt conditions are very interesting, and Movie S5 showing this process is very engaging. The authors should take advice from the journal about the most accessible file formats for movies, as the .mov format is most appropriate for macs, and did not work on my windows laptop (fortunately I also have linux). In support their multi-scale approach, the authors may wish to comment on whether conformational changes this large would have been obtainable over the timescales accessible to atomistic simulation (I suspect not!).

I would like to have a deeper insight into what is happening to give rise to these results. If the DNA is more stable in the bilayer at high salt, then this surely implies that the Martini counterions embed in the lipid bilayer along with the DNA. Is this the case? If yes, how many more counterions are present in high salt compared to low? Where are these counterions preferentially located? Do Martini water particles accompany the counterions? Would the same behaviour be observed if the systems were represented fully atomistically?

The authors comment that more broken base pairs are observed in residues 15-25 in high salt, but I couldn't easily work out which bases these were from the figures. Could the authors indicate where these residues are in one of the structural figures please?

I advise the authors to exert caution when considering the ability of coarse-grained models to describe subtle structural transitions such as B->A-DNA. If the authors have themselves shown, or have evidence from the literature that Martini DNA models can reproduce the B-A transition in DNA, then this should be carefully cited in the paper.

The computational physiology is an interesting addition to the paper. The authors should explain what these measurements are sensitive to, and which aspects of their approach were necessary to reproduce it. For example, how dependent are these measurements on the detailed structures of the pores? What is the key physics that is captured to reproduce the experimental result? This is an important question, because it shows how much effort it required to predict channel conductance in future studies.

There is an interesting statistical physics question concerning the use of ensemble based averages in simulations that complement cryo-EM data. Are the classes observed experimentally thermally accessible from each other? If so, should the simulations have visited all of these classes, rather than sample each class individually? The authors may wish to discuss this.

The authors have the opportunity to refine their language to improve the readability of the manuscript

to a broad audience. For example, instead of simply listing citations to future applications of DNA nanotechnology, the authors could provide a few sentence synopsis on the impact of the field on pure and applied science; e.g in how many of those areas has DNA been used successfully? The first sentence of the abstract is rather strong - what functional materials precisely are only accessible through DNA? Why wouldn't RNA or PNA be feasible? The authors may also wish to tighten their language around the discussion of why "more realistic complex anisotropic environments" are important. This may be true if the DNAs have somehow been introduced into a cell, and clearly for structures embedded in a membrane the lipid bilayer is key. However, not all DNA nanotechnology applications involve lipids and may just be in aqueous solution. Maybe "diverse environments" might be a better wording than "realistic"?

Reviewer #1 (Remarks to the Author):

Reviewer comment:

In this study, Ahmad et al. combine cryoEM imaging and coarse-grained molecular dynamics simulations to characterize in great detail the structure of a six-helix DNA bundle (6HB) in solution. Molecular dynamics simulations are then used to explore the structural consequences of the insertion of the bundle into a lipid bilayer membrane at different salt concentrations. Insertion into the lipid bilayer is mediated by cholesterol membrane anchors positioned close to the duplex nicks.

The results show that in solution the 6HB is rather flexible and assumes conformations that deviate from those expected from design, being typically wider and shorter and with a larger funnel-shaped lumen. Contrarily, insertion of the 6HB into the lipid bilayer results into the reconfiguration of the bilayer into a toroid that laterally squeezes the 6HB, making this adopting a structure that is much closer to the theoretical expected from design. Insertion into the lipid bilayer is also more stable at high salt concentrations.

This is a well-done and accurate simulation study on a relatively simple DNA structure, which essentially confirms and explains the experimental data available from the literature. I do not have special remarks besides a few comments below:

Response:

We thank the reviewer taking time to read and evaluate the manuscript. We also thank the reviewer for the positive assessment. All points of the reviewer have been addressed, and changes in the manuscript are highlighted in yellow.

Reviewer comment:

p. 5: „Our cryo-EM analysis indicated that all five classes feature a funnel-shaped lumen (Figure 3) which is different to the cylindrical lumen of the original 6HB design”

I would recall here again the dimension expected for the lumen (18 Å).

Response:

The sentence has been edited to feature the expected lumen of 18 Å.

Reviewer comment:

“The funnel is narrowest at the top of the 6HB with a diameter of 42 ± 5 Å and narrowest at the bottom with a width of 26 ± 2 Å.” This sentence should be probably reformulated as: “The funnel is narrowest at the top of the 6HB with a diameter of 26 ± 2 Å and widest at the bottom with a width of 42 ± 5 Å”

Response:

Many thanks for highlighting this. This section has been updated with dimensions corresponding to the correct section of the pore lumen. The dimensions have also been updated following harmonization of the analysis method between cryo-EM and MD measurements.

Reviewer comment:

Figure 3B: the y-axis, should be maybe labelled as “Lumen Width” instead of “Lumen Length”?

Response:

Many thanks for highlighting this mistake. It has been rectified.

Reviewer comment:

P6: “...6HB is wider than the indirectly determined lumen width of 18 Å”. I would replace with “...6HB is wider than the nominal value of lumen width, which is 18 Å.”

Response:

The wording has been changed.

Reviewer #2 (Remarks to the Author):

Reviewer comment:

The manuscript from the Howorka group uses cryo-EM combined with coarse-grained molecular dynamics simulations to investigate the structure of a DNA nano-architecture embedded in a membrane. CryoEM studies have revealed a wealth of new information for DNA nanotechnological structures which have been of high scientific impact, as recently reviewed by Pan et al in Small (<https://doi.org/10.1002/sstr.202100053>). My expertise is in computer simulations of proteins and nucleic acids, so I cannot comment on the appropriateness of the experimental methodologies. It is extremely nice to see that computer simulation is playing an increasingly important role in complementing cryo-EM studies, as has been done in the work here. Clearly, the two techniques provide synergistic information that increases insight and understanding.

Response:

We thank the reviewer taking time to read and evaluate the manuscript. We are pleased that the reviewer agrees with our view that combining cryo-EM and simulations synergistically provide insight. The noted review in Small is now cited in the revised manuscript. We would like to point out that the manuscript represents a collaboration of work from three groups, namely those of Coveney, Orlova, and Howorka.

Reviewer comment 1:

There are some aspects of the descriptions of the simulation methodology that it would be helpful to clarify. I found it difficult to understand from reading this version of the paper which simulations (atomistic/coarse-grained/in solution/in bilayer) were performed and why. It might be helpful to include a table summarising this in the Methods or Supplementary Information.

Response: With regard to the inclusion of a table summarising the simulations. The information has been added to the SI, in Table S4. All edited text and newly introduced display items are highlighted yellow in the revised manuscript and SI.

Reviewer comment 2: I was surprised that most of the structural and dynamic analysis reported focused on the coarse-grained simulations and did not include data from the atomistic models. This would be of interest as both length-scales provide complementary information, although more than 50ns of atomistic data would be required (which should not be computationally unfeasible using modern GPU resources). The authors should provide a quantitative description of the helical parameters in the atomistic DNA structures, e.g. using x3DNA or Curves++, to characterise the base-pairing, stacking and backbone conformations in comparison to canonical B-form DNA, and other DNA nanotechnology structures in the literature. If there are significant deviations, then the authors should explain what key inter-atomic interactions are stabilising their structures.

Response: We concur that coarse-graining provides a less-than-ideal description of hydrogen-bonding and other duplex-stabilizing interactions in DNA, and that more detailed atomistic descriptions would be desirable. However, the purpose of this computational study was to showcase the statistical robustness and consequent predictive power of running ensembles of long timescale CG trajectories, which after clustering and CG-to-AA back-mapping, yield representative structures that are in excellent agreement with cryo-EM electron density models. Running such ensembles using atomistic molecular dynamics would be prohibitively expensive, as the number and duration of replicas necessary to produce converged errors is not *a priori* known. Nevertheless, we acknowledge that some analysis of the existing 30 ns – long trajectories of the solvated DNA nanopore in 0.3 M NaCl solutions would strengthen the paper, so we have calculated averaged base-pair and base-step parameters for each of the helices of the 6HB, and listed them in Table S6. We have also discussed the relevance of these parameters in SI section 1.2.6, and present a detailed analysis of the variation of the parameters along the length of the 6HB in Figures S10 and S11, and by doing so we identify the sources of the structural perturbations in the helices of the 6HB. We also provide the information about base-pair breakage in our CG and AA simulations in Table 3. To describe the new data, we have introduced new text at the end of the section ‘Effect of salt concentration on structural dynamics of 6HB in solution’.

In the SI, we have made comparisons to the idealised B-DNA parameters, and the parameters that have been calculated for a series of nicked DNA duplexes and Seeman paranemic crossover DNA nanostructures. Very few published MD studies of DNA nanostructures similar to the 6HB provide these parameters, and even fewer present them in a way that allows direct comparisons to be made. We found more meaningful comparisons could be made by calculating individual parameters for each helix, and making comparisons with simulated structures featuring similar motifs i.e., nicks and crossovers. We have added a sentence summarising the findings from this analysis in the paragraph before Fig. 5, and with text inserted before and after Table 3 of the revised manuscript. In the new passages, we refer the reader also to the SI for more detail (see Figure S5, Figure S9-11, Table S5).

Reviewer comment 3: Having visually inspected the pdb files there was a striking lack of complementary base pair hydrogen bonding and base stacking. The authors should provide a quantitative description of the helical parameters in the atomistic DNA structures, e.g. using x3DNA or Curves++, to characterise the base-pairing, stacking and backbone conformations in comparison to canonical B-form DNA, and other DNA nanotechnology structures in the literature. If there are significant deviations, then the authors should explain what key inter-atomic interactions are stabilising their structures.

Response: A quantitative description of base pairing and other helical parameters have been obtained from our existing all-atom simulations of the DNA nanopore in 0.3 M NaCl solution, as stated in response to the previous point.

Reviewer comment 4: From my reading of the paper, the level of restraints imposed on the coarse-grained DNA simulations as reported in the Supplementary Information is ambiguous. If restraints are indeed needed to maintain the duplex, then how does this affect the dynamics and fluctuations of the DNA reported in Figure 5, and elsewhere? Please could the authors clarify this and explain any potential artifacts that may have been introduced.

Response: We thank the reviewer for raising this very important point. Elastic restraints are employed for MARTINI DNA simulations by default, to maintain the persistence length and double-helix structure (see 10.1021/acs.jctc.5b00286). Two types of elastic restraints can be used, the so called “stiff” and “soft” elastic restraint networks. In our study, we made use of the former – which applies restraints to all pseudo-atom beads within the nucleotides, thus restricting the flexibility of the bases as well as the backbone. Naturally, this means that the RMSF (see plots in Figure 5) of the nucleotides is systematically underestimated, and the restricted flexibility of the bases may also contribute to the exaggerated percentage of base pair breakage that is seen in the CG models. We have added a comment clarifying this point in the figure legend of Figure 5, and added a figure demonstrating the difference in fluctuations between the AA and CG models to the supplementary information (the new Figure S8).

Reviewer comment 5: The results showing that the DNA-nanostructure is unstable in the lipid bilayer under low salt conditions are very interesting, and Movie S5 showing this process is very engaging. The authors should take advice from the journal about the most accessible file formats for movies, as the .mov format is most appropriate for macs, and did not work on my windows laptop (fortunately I also have linux). In support their multi-scale approach, the authors may wish to comment on whether conformational changes this large would have been obtainable over the timescales accessible to atomistic simulation (I suspect not!).

Response: We agree that these events would have almost certainly not been observed if an atomistic MD protocol was employed for this investigation. We thank the reviewer for highlighting this point, and we have added a sentence in the first paragraph of section entitled ‘Effect of salt concentration on structural dynamics and stability of membrane-inserted 6H’. The movies are also provided in format m4v compatible with PCs.

Reviewer comment 6: I would like to have a deeper insight into what is happening to give rise to these results. If the DNA is more stable in the bilayer at high salt, then this surely implies that the Martini counterions embed in the lipid bilayer along with the DNA. Is this the case? If yes, how many more

counterions are present in high salt compared to low? Where are these counterions preferentially located? Do Martini water particles accompany the counterions? Would the same behaviour be observed if the systems were represented fully atomistically?

Response: We appreciate the reviewer's interest in this part of the manuscript, and agree that more analysis of the role of water and ions would bolster it. We have included an analysis of the water and ion densities across the simulation box, which illustrates the sharp increase in the density of ions from the edges of the simulation box (bulk solution around the membrane) to the centre of the box where the 6HB pore's lumen is located, suggesting that positive ions aggregate inside the lumen. In addition, this increase in density is higher in magnitude in 1.0 M NaCl than in 0.3 M NaCl. We have also provided raw number of ions residing in the pore lumen (Table S5). To address the reviewer's question about whether or not the same behaviour would be seen in AA simulations, we briefly discuss the artefacts that the larger size of MARTINI particles introduce in radial distribution functions between MARTINI ions and charged biomolecules in the paragraph before Fig. 6. We conclude that the published evidence suggests that, despite subtle difference between AA and CG rdfs, local ion densities in the proximity of charged polymers are generally reproduced well by the MARTINI model with polarizable water.

Reviewer comment 7: The authors comment that more broken base pairs are observed in residues 15-25 in high salt, but I couldn't easily work out which bases these were from the figures. Could the authors indicate where these residues are in one of the structural figures please?

Response: Information about the base pair positions has been added to Figure 5a.

Reviewer comment 8: I advise the authors to exert caution when considering the ability of coarse-grained models to describe subtle structural transitions such as B->A-DNA. If the authors have themselves shown, or have evidence from the literature that Martini DNA models can reproduce the B-A transition in DNA, then this should be carefully cited in the paper.

Response: We have edited the manuscript to clarify the location of broken base pairs (Table 3, Figure S10, S11, Table S5). We acknowledge that CG force fields cannot describe beta-to-alpha transitions; we simply aimed to indicate that there is a loss of the idealised beta structure in some parts of the nanopore. Additional text about the broken base pairs has been added at the end of section entitled 'Effect of salt concentration on structural dynamics of 6HB in solution'.

Reviewer comment 9: The computational physiology is an interesting addition to the paper. The authors should explain what these measurements are sensitive to, and which aspects of their approach were necessary to reproduce it. For example, how dependent are these measurements on the detailed structures of the pores? What is the key physics that is captured to reproduce the experimental result? This is an important question, because it shows how much effort is required to predict channel conductance in future studies.

Response: To address this point, we have edited the manuscript to provide more information on the computational electrophysiology (see end of first paragraph in Computational electrophysiological analysis of membrane-inserted 6HB). We anticipate that the lumen diameter and properties affecting the hydrophobicity of the material will be among the parameters that dominate the uncertainty in our simulations. A fully systematic sensitivity analysis and study of the sources of uncertainty in the model is beyond the scope of this work. The state of the art in uncertainty quantification, which is new in the field of particle-based modelling and simulation, was recently assessed by one of us (PVC) in a theme issue [Coveney et al. (eds) <https://doi.org/10.1098/rsta.2020.0409>: please note that the reference is intended to cover uncertainty quantification (UQ) in science as a whole, but there is one paper within the issue which is dedicated to UQ in classical molecular dynamics, DOI:10.1098/rsta.2020.0082]. It requires one to probe all the parameters employed in the model through a series of computational campaigns, as well as the dependence of the quantities of interest on the random number seeds used in the code [DOI:10.1021/acs.jctc.1c00526.] We also refer the reviewer to the following papers (which have been cited in the manuscript) for more information about the Computational Electrophysiology protocol: 10.1016/j.bpj.2011.06.010, 10.1016/j.bbamem.2016.02.006W

Reviewer comment 10: There is an interesting statistical physics question concerning the use of ensemble-based averages in simulations that complement cryo-EM data. Are the classes observed experimentally thermally accessible from each other? If so, should the simulations have visited all of these classes, rather than sample each class individually? The authors may wish to discuss this.

Response: All of these cryo-EM states are thermally accessible from one another. An ensemble of microstates initiated in the manner we describe reaches local equilibrium on a time scale of a few tens of microseconds. Thus we attain a macrostate comprising numerous cryo-EM states that emerge on such a timescale. To capture all of them, one would need to run a much larger ensemble for significantly longer (possibly microseconds). This is nevertheless far shorter in elapsed physical and computer wall time than attempting to observe all such states within a single trajectory. This would require something of the order of a Poincaré recurrence time, which is extremely long and totally out of reach on any present day computer (P. V. Coveney and S. Wan, "On the calculation of equilibrium thermodynamic properties from molecular dynamics", *Phys. Chem. Chem. Phys.*, 2016, 18, 30236-30240, DOI: 10.1039/C6CP02349E).

Reviewer comment 11: The authors have the opportunity to refine their language to improve the readability of the manuscript to a broad audience. For example, instead of simply listing citations to future applications of DNA nanotechnology, the authors could provide a few sentence synopsis on the impact of the field on pure and applied science; e.g in how many of those areas has DNA been used successfully? The first sentence of the abstract is rather strong - what functional materials precisely are only accessible through DNA? Why wouldn't RNA or PNA be feasible? The authors may also wish to tighten their language around the discussion of why "more realistic complex anisotropic environments" are important. This may be true if the DNAs have somehow been introduced into a cell, and clearly for structures embedded in a membrane the lipid bilayer is key. However, not all DNA nanotechnology applications involve lipids and may just be in aqueous solution. Maybe "diverse environments" might be a better wording than "realistic"?

Response: In the revised introduction, the first paragraph explains more the applications of DNA nanotechnology. The first sentence of the abstract is also edited to tone down the too strong claim. In nucleic acid nanotechnology, DNA is preferred over RNA and particularly PNA as they are much more expensive to produce. In addition, PNA with its lack of negative charges often suffers from solubility issues. The term 'realistic environments' has been replaced with 'diverse environments'. Furthermore, the manuscript has been edited in several other passages to improve readability.

Reviewers' Comments:

Reviewer #1:

Remarks to the Author:

The authors fully addressed my few questions.

I am happy to see that the present version of the manuscript even further gained in accuracy and details, particularly in response to the points raised by the other reviewer.

Altogether, this study adds fundamental knowledge to the cryo EM/DNA nanotechnology community and it does in a very tidy and clear way. I therefore fully support publication of this work in Nature Communication.

Reviewer #2:

Remarks to the Author:

The referees acknowledge that the authors have made a strong effort to address their comments. The authors have provided the structural analysis of the DNA simulations requested in their revised manuscript. This indicates that the DNA structures reported lack the structural integrity required for them to be considered reliable. For example, in Figure 2 the DNA structures look significantly distorted. Table 1 reports RMSD values as large as 13.5 angstroms, and Table 3 shows that there is large disruptions in the complementary hydrogen bonds. There is a strong possibility that this is due to the coarse-grained force-field applied, which may not cope well with the subtleties of highly charged DNA inserted into a lipid bilayer. The authors have not been able to demonstrate that the simulations they report are reliable or state of the art in their analysis, and they need to do this before they publish their simulation data.

Reviewer #3:

Remarks to the Author:

The manuscript combines cryo-EM and MD simulation to reveal the structure and dynamics of a DNA nanoarchitecture (6HB) under different conditions. The authors identified non-hexagonal and distorted structure outside lipid bilayers by cryo-EM and simulation. And MD simulation indicate that in lipid membranes, 6HB restores barrel pore structure due to the compression from lipids. This is a very interesting observation. I was wondering if the authors can confirm this phenomenon using cryo-EM. Now, reconstructing membrane proteins in lipid nanodiscs is a very common method. Have the authors tried to use lipid nanodisc to prepare the cryo sample to study 6HB structure in lipid environment?

Reviewer #1 (Remarks to the Author):

The authors fully addressed my few questions.

I am happy to see that the present version of the manuscript even further gained in accuracy and details, particularly in response to the points raised by the other reviewer. Altogether, this study adds fundamental knowledge to the cryo EM/DNA nanotechnology community and it does in a very tidy and clear way. I therefore fully support publication of this work in Nature Communication.

Reply: We thank the reviewer for the positive comments of our revised manuscript and the recommendation to accept our study for publication in Nature Communications.

Reviewer #2 (Remarks to the Author):

The referees acknowledge that the authors have made a strong effort to address their comments. The authors have provided the structural analysis of the DNA simulations requested in their revised manuscript. This indicates that the DNA structures reported lack the structural integrity required for them to be considered reliable. For example, in Figure 2 the DNA structures look significantly distorted. Table 1 reports RMSD values as large as 13.5 angstroms, and Table 3 shows that there is large disruptions in the complementary hydrogen bonds. There is a strong possibility that this is due to the coarse-grained force-field applied, which may not cope well with the subtleties of highly charged DNA inserted into a lipid bilayer. The authors have not been able to demonstrate that the simulations they report are reliable or state of the art in their analysis, and they need to do this before they publish their simulation data.

Reply:

We thank the reviewer for acknowledging our strong effort to address the referee comments. In the new comments, the reviewer states the DNA simulations lack structural integrity. Figures 2 and 3 as well as Table 1 are cited to support the claim that the used coarse-grained force fields do not cope well with the details of the DNA structures. In reply, we note that there are significant misunderstandings by the reviewer. Figures 2 and 3 and Table 1 report on experimental structural data obtained from cryo-EM studies. The cited data have nothing to do with computational simulations and no force fields were applied for these set of data. The distortion the reviewer observed in Figure 2 is precisely what we intended to display: without membrane, the DNA nanostructure is distorted from the designed structure in solvent. Once inserted, the expected DNA helix bundle structure is restored.

Reviewer #3 (Remarks to the Author):

The manuscript combines cryo-EM and MD simulation to reveal the structure and dynamics of a DNA nanoarchitecture (6HB) under different conditions. The authors identified non-

hexagonal and distorted structure outside lipid bilayers by cryo-EM and simulation. And MD simulation indicate that in lipid membranes, 6HB restores barrel pore structure due to the compression from lipids. This is a very interesting observation. I was wondering if the authors can confirm this phenomenon using cryo-EM. Now, reconstructing membrane proteins in lipid nanodiscs is a very common method. Have the authors tried to use lipid nanodisc to prepare the cryo sample to study 6HB structure in lipid environment?

Reply:

We thank for the reviewer for the positive assessment of our revised manuscript. Lipid nanodiscs are certainly an interesting option. While common, setting up the nanodisc system would be beyond scope of the current work. However, we have edited the results and discussion to clearly state that nanodiscs have not been used but may be applied in future studies to experimentally examine the structure of DNA nanopores in membranes.